# The Clinical Benefits of a Dynamic vs. Static Component as Part of a Comprehensive Warm-Up for Recreational Sports Players with Clinical Histories of Hamstring Injuries: A Randomized Clinical Trial

**DOI:** 10.3390/ijerph20010744

**Published:** 2022-12-31

**Authors:** Javier Gutierrez-Coronado, Laura López-Bueno, María de los Angeles Cardero-Durán, Manuel Albornoz-Cabello, Jose Vicente Toledo-Marhuenda, Sergio Hernández-Sánchez, Lirios Dueñas, Elena Marques-Sule, Antoni Morral, Luis Espejo-Antúnez

**Affiliations:** 1Department of Medical-Surgical Therapeutics, Faculty of Medicine and Health Sciences, University of Extremadura, Av. Elvas, s/n, 06006 Badajoz, Spain; 2Department of Physiotherapy, Faculty of Physiotherapy, University of Valencia, Gascó Oliag Street, 5, 46010 Valencia, Spain; 3Department of Physical Therapy, Faculty of Nursing, Physical Therapy and Podiatry, University of Seville, C/Avicena, 6, 41009 Seville, Spain; 4Department of Pathology and Surgery (Area of Physiotherapy), Medicine Faculty, Miguel Hernández University, Ctra Alicante-Valencia Km 8,7—N 332, 03550 Alicante, Spain; 5Physiotherapy in Motion, Multi-Speciality Research Group (PTinMOTION), Department of Physiotherapy, Faculty of Physiotherapy, University of Valencia, 46010 Valencia, Spain; 6Blanquerna School of Health Sciences, Ramon Llull University, Carrer de Claravall, 1, 3, 08022 Barcelona, Spain

**Keywords:** static warm-up, dynamic warm-up, recreational sport players, hamstring injury

## Abstract

Background: Few previous studies have analyzed the effects of certain specific static and dynamic warm-up components on recreational sports players with a previous hamstring injury. Therefore, the aim of this study was to analyze changes in some modifiable and external risk factors after (immediately and in a follow-up assessment after 10 min) a static or dynamic warm-up program on recreational sports players with a previous hamstring injury. Methods: A total of 62 participants were randomized into 2 groups: static warm-up (SW) (n = 31) or dynamic warm-up (DW) (n = 31). Range of movement (RoM), perceived pain, the pressure–pain threshold, and joint position sense were assessed at baseline, immediately after the intervention and 10 min afterwards. The intervention for the SW (hot pack procedures in both hamstring muscles) lasted 20 min. The DW intervention consisted of a running exercise performed on a treadmill for 10 min. Results: Both groups showed statistically significant changes (*p* ≤ 0.05) in the primary outcomes (perceived pain and the pressure–pain threshold) at the three measurement times (this was also true for RoM for the SW group, with statistically significant differences only between times from the baseline to the 10-min follow-up; *p* ≤ 0.05, d = 0.23). The intra-group secondary outcome showed no statistically significant changes (*p* > 0.05) in both groups (except for the period from the baseline–immediately after in the DW group; *p* ≤ 0.05, d = 0.53). The comparison between groups showed no statistically significant differences for any of the variables analyzed. (*p* ≥ 0.05). Conclusion: The present findings suggest that both specific warm-up modalities seem to positively influence perceived pain on stretching and the pressure threshold; however, the significant reduction in the joint repositioning error and the larger effect sizes observed in the DW group suggest that this method has a greater beneficial impact in recreational sports players with clinical histories of hamstring injuries.

## 1. Introduction

Any activity that increases one’s body temperature by few degrees Celsius is defined as a “warm-up” [1]. Specifically, a warm-up in a sport is defined as a period of preparatory exercise used to enhance subsequent competition or training performance [2]. It is common in many different sports disciplines [3,4] and has proven beneficial effects for preventing sports-related injuries and reducing their number and severity [1,5,6].

A warm-up can be classified in two modalities: dynamic warm-up (DW) or a static warm-up (SW). During a DW, body temperature is elevated due to the energy released from contracting muscles. This type of warm-up helps to increase blood flow, and also increases baseline oxygen consumption and leads to the breaking of actin and myosin bonds, which improves flexibility [1].

During an SW, external methods such as thermotherapy, diathermy, and hot packs are used to raise the tissue or body temperature, relieving pain and increasing the elasticity of connective tissues [7]. This can help to increase muscle flexibility and the joint range of movement (RoM). In addition, it helps to increase oxygen release from hemoglobin and myoglobin, which in turn increases the metabolism of the energy system and decreases the peak tension time in muscles [7]. In this sense, superficial heat in the form of moist hot packs has traditionally been used, combined with a stretching program, to increase the flexibility of muscles prior to physical activity [8]. However, warm-up routines in sports include different forms and modalities, isolated or combined (running, exercises, static or dynamic stretching, dynamic movements, heat modalities, etc.). This makes it difficult to agree on their supposed clinical benefits. Currently, studies on the impact generated by specific warm-ups and their components are limited, and certainly controversial.

In this sense, it has been suggested that dynamic heating by means of dynamic stretching produces superior physiological performances, compared to static stretching, due to the increase in corticospinal excitability [9]. However, O’Sullivan et al. (2009) reported that static stretching followed by an aerobic warm-up obtained superior improvements over dynamic stretching in terms of range of motion [3]. Recently Blazevich et al. (2018) reported it to be unlikely that the inclusion of short-duration static or dynamic stretching in a global warm-up could affect sports performance when this was performed as part of a comprehensive physical preparation routine [10].

Gogte et al. (2017) studied the differences between some dynamic warm-up components (e.g., leg press exercises and use of a stationary bicycle) and the isolated application of moist heat [1]. The lack of significant differences between groups for sports performance variables such as the Vertical Jump Test (VJT) or the Star Excursion Balance Test (SEBT) affirms the need to analyze the clinical effects on perceived pain and range of movement (RoM), among other variables, of certain components of warm-ups (DW or SW) that are incorporated into the specific warm-up of a sport or physical activity.

The effect of warming up prior to engaging in sports activities has mainly been studied in the lower limbs, due to their high injury incidence rate. In particular, the incidence of hamstring injuries is estimated to be above 6 players per season with an average of 18 days and 3–3.5 matches missed per hamstring strain [11]. In addition, a third of these injuries will recur during the first two weeks after the start of competitive sports activity [12].

Despite the existence of studies that analyze the effects of certain components (e.g., heat modalities, education, stretching, etc.) within a warm-up routine [4,6,8,13,14], to our knowledge, studies designed to evaluate the impact of specific components of the static and dynamic warm-up routines of recreational sports players with clinical histories of hamstring injury are almost non-existent [3].

Therefore, the aim of this study was to examine and compare the clinical benefits of a treadmill-based dynamic warm-up versus a static warm-up (consisting of the use of hot packs) in recreational sport players with previous hamstring injuries in the dominant lower limb.

## 2. Materials and Methods

### 2.1. Design and Setting

A randomized single blind trial was conducted to compare the effects derived from the application of two different warm-up modalities on the clinical parameters of recreational sports players with previous hamstring injuries.

### 2.2. Participants

A total sample of 62 recreational sport players, all of them undergraduate students, voluntarily participated in the study. The inclusion criteria were as follows: aged from 18 to 27, currently physically active and practicing a recreational sporting activity for at least 5 h per week during the last 2 years, and having suffered a hamstring injury classified as 2b according to the consensus of Munich [15] (diagnosed by a doctor) in this time, but not within the last month. Finally, participants were required to have a straight leg raise test of less than 80° at the start of the study. Hamstring injury is defined as a wound that causes the subject to stop running, or suffer from pain or stiffness during the 10 subsequent days while running. Participants were excluded if they had a history of vestibular disorders, consumed drugs that could alter body balance, or had participated in an organized hamstring muscle stretching program. Participants whose sports performance was part of their profession were also excluded.

The sample size calculation was computed (software G*Power version 3.1.9.7-Heinrich-Heine University, Düsseldorf, Germany) based on data from a previous pilot study with 10 participants; the inclusion/exclusion criteria were as mentioned before. Considering a one-tail hypothesis, for ANOVA, repeated measures between interaction (2 × 3, to evaluate group × time interactions), an alpha value of 0.05, a desired power of 80%, and an effect size (η^2^ = 0.3), 62 participants were required in the total sample size. All subjects were screened for inclusion and exclusion criteria and were ultimately selected for the study. The allocation of the subjects is explained in the flowchart (Figure 1).

After the baseline assessment, participants were randomly assigned into two groups. Randomization was performed by asking the participant to pick up an opaque and sealed envelope before the baseline data collection (inside was written “1”—corresponding to the static warm-up group; “2”—corresponding to the dynamic warm-up on a treadmill). A researcher, who was aware of the study design, conducted enrollment and group assignment. The outcome assessors were blinded to group allocation. Participants were blinded to the group assignment and were assessed simultaneously from the beginning of the procedure to its end.

### 2.3. Intervention

All subjects were assessed individually, in the same order and across three different moments, as proposed by Blazevich et al. [10]: before the assigned warm-up, immediately after the warm-up, and 10 min after ending the warm-up.

All data were collected in the same room, at a temperature of 22–25 °C, with the participants wearing comfortable clothes (t-shirts and shorts), and considering the dominant lower limb. Height and weight measurements were obtained using a standard scale and stadiometer (Seca 285, Seca, Birmingham, UK).

All subjects started with a conventional pre-activity routine consisting of a low-intensity aerobic warm-up (<60% maximum heart rate) on a cycle ergometer (5-min). Subsequently, a routine of 4 exercise components (30 s per exercise) was applied (Figure 2). The exercises consisted of: (a) leg extension in isometric contraction (30 s); (b) active mobilization with knee flexion and extension; (c) neurodynamic tensioner with ankle dorsiflexion; (d) neurodynamic slider with active knee extension. The indicated rest period after the completion of each component was 1 min. These exercises have been shown to be beneficial in reducing muscle injuries and increasing joint RoM, without effecting sporting performance [13,14,16,17].

After this, the subjects belonging to group 1 and group 2 were instructed to perform different warm-ups, as explained below:

**Static Warm-Up method (SW):** Subjects were placed in a prone position, with their limbs uncovered. They received 3 standard-size hot packs (The Netherlands. Enraf Nonius^®^). Hot packs (gel fillers) that had previously been heated up to 60 °C in a microwave were adjusted to completely cover the entire hamstring muscle in both legs (Figure 3). The temperature of the hot packs was confirmed with a thermometer, after several tests to determine the exact time needed for heating in the microwave (Taurus^®^,Taurus, Cataluña, Spain). The duration of this local application was 20 min. According to previous studies, this period is considered long enough to produce an increase in muscle temperature [1,18]. Participants were alerted that the sensation of heat caused by the hots packs was not going to exceed their own tolerance. In addition, the intervention was applied to both lower limbs to avoid bias between interventions.

**Dynamic warm-up method (DW):** Considering that running is the aerobic component most frequently included in warm-up programs [5], the subjects assigned to this group were instructed to run on a treadmill for 10 min at an intensity equivalent to 70% of their maximum heart rate. The pace had to be continuous and regular, without an imposed cadence, to avoid muscle fatigue [19].

### 2.4. Outcomes

#### 2.4.1. Primary Outcomes

**Knee extension range of movement (RoM):** An Active Knee Extension (AKE) test was used to assess the hamstring extensibility and the subject’s specificity to identify the possible changes induced by the intervention [20]. The subject was lying in supine position with hip and knee at 90° flexion, keeping the tibia in a horizontal position and the ankle in a neutral position, while maintaining contact with the indicated points of reference. The participant’s leg was actively extending to induce extension at the knee with minimal hip rotation until the participant verbally indicated that the point of discomfort was reached. The point of discomfort was described to the participants as the point at which they felt the onset of uncomfortable tension in the hamstrings. The measurement was performed three times, and the mean of the three values was calculated for the analysis. Exact knee angles were determined by computer analysis of the photographs using Posture Assessment Software (SAPO^®^). This method has previously shown excellent reliability (intraclass correlation coefficient = 0.96) in the measurement of knee RoM [21]. The knee angle was determined as the average of five consecutive photographs of each position. The mean values were used for the statistical analysis.

**Stretching Tolerance:** Stretching tolerance was assessed using the Visual Analogic Scale (VAS). Each subject rated the intensity of perceived pain at maximum extensibility in the AKE test on a straight horizontal line of 100 mm length, where 0 was no pain and 100 was maximum pain [22].

**Pressure–Pain Threshold (PPT)**: The PPT is defined as the amount of pressure applied at which the pressure sensation first changes to pain [23]. This was assessed using a mechanical pressure algometer (model FPX 25, Wagner Instruments, Greenwich, CT, USA) with a 1 cm^2^ area contact head. The reliability of pressure algometry has been found to be high (intraclass correlation coefficient = 0.91 (0.82–0.97)) [24]. With the participant in a prone position, the pressure–pain threshold of the central trigger point (TrP) of the biceps femoris muscle (dominant lower limb) was evaluated. The pressure algometer was placed as Travell and Simons [25] describe. A gradual and continuous pressure was applied perpendicular to the biceps femoris, approximately 1 kg/cm^2^/sec, registering the necessary pressure to provoke a sensation of moderate pain [26].

#### 2.4.2. Secondary Outcome

**Joint Position Sense**: This is defined as the precision capacity of an individual to replicate a joint angle in each position. It constitutes an important factor of proprioception to promote functional stability in sports practice. Four reflective sticker markers were fixed to the skin: (1) over the apex of the greater trochanter; (2) at the iliotibial tract, level with the posterior crease of the knee, when flexed to 80°; (3) on the neck of the fibula; and (4) on the prominence of the lateral malleolus. The subject remained in a prone position with the leg bent at 90° of knee flexion in the initial position (measured with a goniometer). After that, the subject bent his leg until reaching 45° of knee flexion (in the direction of extension). The subject had to memorize and hold both positions to ensure his capacity for joint repositioning. The established time to guarantee the proprioception acuity was 6 s, both for the initial position (90°) and for the final position (45°) [27]. The resulting measurement was obtained as the average of three completed attempts.

### 2.5. Statistical Analysis

A descriptive analysis of the samples’ anthropometric variables was completed. The normality of data distribution was tested with the Shapiro–Wilk Test. The data are reported as mean ± SD or median (interquartile range). The Student’s paired *t*-test and the Wilcoxon signed-rank test were performed for comparisons within groups, for data with normal and non-normal distribution, respectively.

Friedman’s two-way analysis was used to test differences between the three data collection points (pre-test, post-test, and 10 min afterwards), for data from all variables (significant interaction effect). A repeated-measures ANOVA was used to examine the effects of the interventions on the AKE test, stretching tolerance, the pain–pressure threshold, and joint position sense, between groups over time (group × time). Post-hoc comparisons were performed using Bonferroni Tests. Furthermore, the effect size was calculated through Cohen’s d coefficient and interpreted as small (d = 0.2), medium (d = 0.5), or large (d > 0.8) [28]. The significance level was established at *p* < 0.05. Data analysis was conducted using the statistical software IBM SPSS Statistics version 20.0 (SPSS Inc., Chicago, IL, USA).

### 2.6. Ethical Considerations

The research protocol was approved by the Institutional Review Board (Research Ethics Committee) with registration number 37/2013. The study protocol was conducted according to the Declaration of Helsinki and was registered at www.clinicaltrials.gov (accessed on 9 October 2022). (NCT03444285). Prior to data collection, all participants were informed about the study procedures. Written informed consent was obtained from all participants before the study began.

## 3. Results

Baseline descriptive characteristics of the sample, both demographic and clinical, are shown in Table 1.

The intragroup comparisons of the static warm-up (SW) at different times are shown in Table 2. Immediate statistically significant changes (*p* < 0.05) were observed in perceived pain (VAS) (13.9 ± 13.5; *p* < 0.001) and the PPT (*p* < 0.001 Dif: −1.1 ± 1.2 kg/cm^2^). The parameters RoM (*p* = 0.079) and JPS (*p* = 0.153) did not show significant improvements (*p* ≥ 0.05). Moreover, 10 min after the intervention, statistically significant differences were maintained for VAS: *p* = 0.008; PPT: *p* = 0.020), obtaining a statistically significant RoM gain at 10 min (*p* < 0.05). Effect sizes were low–moderate (VAS: d = 0.41; PPT: d = 0.32, RoM: d = 0.23) (Table 2). No statistically significant changes were observed for the rest of the parameters, except in the comparison of the initial evaluation and 10 min after the intervention (2.1 (0.4–3.7); *p* = 0.015).

In addition, the results obtained in the dynamic warm-up (DW) group were similar to those obtained in the SW group. Immediate statistically significant changes were obtained in stretching tolerance (14.84 ± 14.7 mm; *p* < 0.001), PPT (1.69 ± 1.3 kg/cm^2^; *p* < 0.001) and JPS (1.5 (0.3–2.6) *p* = 0.016). The differences observed were maintained 10 min after the intervention, except for the JPS. Effect sizes were moderate–large (VAS: d = 0.57; PPT: d = 0.72) (Table 2).

Table 3 shows the measurements taken immediately after and in the 10-min follow-up, as well as the between-group mean differences. Differences that were not statistically significant were found between the static and dynamic specific warm-up intervention groups in range of movement (RoM: F_1,60_ = 1.51 (*p* = 0.223) η^2^ = 0.025); stretching tolerance (VAS F_1,60_ = 0.531 (*p* = 0.469)) η^2^ = 0.009; the pressure–pain threshold (PPT: F_1,60_ = 084 (*p* = 0.363) η^2^ = 0.014); and Joint Position Sense (JPS: F_1,60_ = 0.088 (*p* = 0.768) η^2^ = 0.001).

## 4. Discussion

The aim of this study was to analyze the short term effects (immediately and in a 10 min follow-up assessment) on clinical parameters of a static or dynamic specific warm-up program in recreational sports players with a previous hamstring injury.

The primary goal of a hamstring rehabilitation program is for the athlete to return to playing sport at their prior level of performance, with minimal risk of injury recurrence [29]. In this regard, the best-available evidence suggests clinical and performance benefits following warm-up procedures or the reduction of muscle cooling [30,31]. However, the benefit derived from each component (static vs. dynamic) is still up for debate. The results of the present study could contribute to the selection of the type of warm-up undertaken in the daily sports practice of recreational sports players with history of hamstring injury.

The main findings of this study were that interaction effects between specific warm-up procedures for both primary outcomes were maintained in the 10-min follow-up assessment. This was the main effect of the warm-up procedures, and no statistically significant differences were shown between the SW and DW groups in terms of RoM and improved perceived pain intensity. Nevertheless, a considerable decrease in the joint repositioning error and bigger effect sizes were seen in the DW group, suggesting that running has superior clinical advantages as compared to using hot packs for the purpose of warming up as a recreational sports player with a clinical history of hamstring injury.

Regarding RoM, controversies exist regarding the duration and intramuscular temperature that must be reached to report clinical benefits. Ostrowski et al. (2017) [32] reported the need to increase the intramuscular temperature by 3–4° Celsius (Cs), while Sawyer et al. [33] did not increase the temperature by more than 0.4° Cs after 20–25 min of application. Despite this, previous studies show improvement after the local application of heat modalities on the calf [8,31,34]. The statistically significant changes observed after 10 min in the SW group (Table 2) could be due to the acute effect on viscoelastic properties after local application of hot packs to the muscle. [31]. Funk et al. [35] showed similar increases in RoM in recreational sport players with hamstring shortness after the use of a moist heat pack during 20 min (1.24° after the test and 2.62° after 8 min). However, the participants in the present study were physically active (asymptomatic at the time of the measurements), and RoM improvements were lower than the minimal detectable change (MCD_95_ = 12°) found for subjects with the same characteristics (MCD_95_ = 12°) [36] to reduce its clinical benefits (SW group: 2.1° (0.4–3.7); d = 0.23) (Table 2). Likewise, the changes observed in both groups are similar to the error of the standard measurement (SEM) from the active extension knee test: from 1.04° to 2.16 for subjects with short hamstring muscles [37]. In this sense, Cosgray et al. [8] did not find significant changes in RoM after two kinds of thermotherapy with a control group comparison in a sample with 30 male university students.

On the other hand, the significant improvement in the stretching tolerance observed immediately after the intervention in both groups could be related to three mechanisms. First, there are neural adaptations derived from the pre-activity routine with exercise components of soft tissue stretching (muscular and neural) (Figure 2). Nerve adhesions in the hamstring may cause abnormal mechanosensitivity of the sciatic nerve, which can contribute to perceived pain [38]. Hatano et al., 2019 [39] hypothesized that the primary mechanism for this improvement is the decrease in the excitability of ventral horn cells after stretching exercises. Second, acute reductions in muscle and tendon stiffness [16] improve the sensitivity of the sciatic nerve to movement [40]. Third, vascular morphological and physiological changes occur after the application of a 60° hot-pack [18]. Recent studies have shown the clinical benefits on perceived pain both of applying hot packs locally after exercise [41] and as a component of a pre-activity warm-up routine [8,35].

Regarding PPT, the intragroup results showed statistically significant increases for both groups, which were maintained after 10 min in a follow-up examination (Table 2). No statistically significant changes were observed between the groups (Table 3). There are few studies that analyze mechanosensitivity by means of algometry following warm-up procedures. McCray and Patton [26] compared the effects of moist hot packs and shortwaves on the TrPs of various muscles (thoracic lumbar and gluteal), showing beneficial changes in both interventions. Studying a group of participants of similar age and BMI to our study, Vaegter et al. (2018) [42] obtained similar increases in the quadriceps muscle after comparing a warm-up program (15 min of cycling) to inactivity (a rest period).

In both groups, the mean differences in stretching tolerance and PPT reported immediate large–moderate effect sizes (SW group d = 0.82 and d = 0.6, respectively; DW group d = 0.9 and d = 1.01, respectively) and after 10 min of follow-up (SW group d = 0.41 and d = 0.32, respectively; DW group d = 0.57 and d = 0.72, respectively) (Table 2). It is possible that the activation of central mechanisms of endogenous pain inhibition mediate the improvement of both stretching tolerance and PPT after an SW and after a DW. Additionally, the combination of the pre-activity warm-up routine together with the static (hot packs) and dynamic (treadmill running) component could minimize the loss of muscle tissue temperature, reducing peripheral nerve excitability and, consequently, perceived pain [43].

With respect to the JPS at a target angle of 45°, statistically significant changes with a moderate effect size were obtained in the immediate post-test for the DW group (d = 0.53). Active heating, such as that achieved by running, generally produces several benefits such as increased temperature and muscle energy metabolism. This could perhaps increase the activity of muscle spindles and, subsequently, the afferent proprioceptive feedback [44]. Despite these results, the lack of significant interaction (Table 3) suggests limited clinical benefits, in contrast to previous studies. Daneshjoo et al., 2012, reported 45 statistically significant changes at a target angle after the application of two popular warm-up programs: the warm-up FIFA 11+ program (2.8 (24.4 to 21.2); 2.8%) and the Harmoknee program (3.1 (25.2 to 20.9); 3%) [44]. Walsh also achieved superior improvements in knee joint position sense measurement when combined with cycling (15 min) followed by static or dynamic stretching, compared to cycling alone [45]. The fact that these interventions were carried out over a longer period of time (three sessions per week for eight weeks) [44], as compared to the present study, could influence these differences. Equally, the stretching exercise could be applied to agonist and antagonist musculature (quadriceps/ischiotibial) as part of the warm-up procedure and with longer durations (90 s, 3 series of 12 repetitions) [45] than those proposed for each fiscal year in this study (30 s) (Figure 2).

On the other hand, our results are consistent with the assertions made by Proske (2019) [46] on the direction of the effects of exercise on the knee repositioning error. In this revision study, it was indicated that, after exercise, the subjects felt the muscle to be longer; therefore, the direction of the angular error in positional sensation occurred in the extension movement. In our study, running provoked eccentric activity (muscle lengthening) on the hamstring muscles, the direction of the observed angular error being consistent with these statements.

### 4.1. Limitations

The study design did not allow for a control group, meaning that the relative contribution of each component was not established. Additionally, in view of the results obtained, future studies combining both interventions against a control group are necessary. The follow-up period was extended to 10 min after the procedure. Future studies with longer follow-up times are necessary to determine the medium-term extent of the results. Since proprioception involves peripheral and central components, the observed improvement in JPS may be due to other factors that could not be controlled for in the present study. Likewise, the knee joint position sense measurement was evaluated only in the open kinematic chain. Other factors, such as the sport discipline, the competitive level, BMI, or unmodifiable factors, could also affect the results.

### 4.2. Clinical Implications

The main finding of this study indicates that a comprehensive warm-up program for the lower limbs (with static and dynamic components) improves hamstring stretch tolerance and the pain threshold to pressure on the hamstring muscles of recreational sports players with clinical histories of hamstring injuries. A higher effect size was shown in the “dynamic warm-up” group for these variables, and statistically significant differences were observed in the knee joint position sense measurement; this suggests that running produces better clinical benefits than applying hot packs. Nevertheless, we agree with Magalhães et al. [4] that caution is needed when extrapolating the results found in the laboratory to the warm-up before sports practice. These results could have relevant clinical implications in muscle injuries, as established in the consensus of Munich [14], which have been little explored to date.

## 5. Conclusions

The present findings suggest that both specific warm-up modalities (i.e., hot packs and treadmill running) seem to positively influence perceived pain on stretching and the pressure threshold. However, the significant reduction in the joint repositioning error and the larger effect sizes observed in the dynamic warm-up group suggest that running has a greater beneficial impact than applying hot packs in recreational sports players with clinical histories of hamstring injuries, although additional research is warranted.

## Figures and Tables

**Figure 1 ijerph-20-00744-f001:**
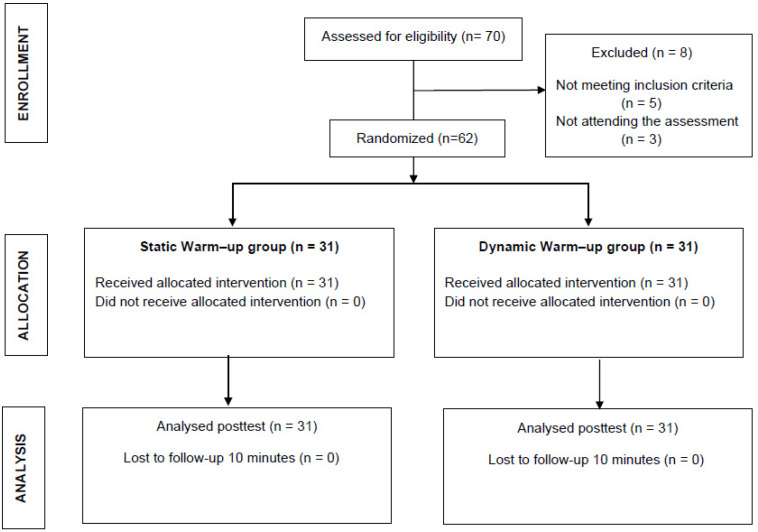
Flow diagram of participant recruitment.

**Figure 2 ijerph-20-00744-f002:**
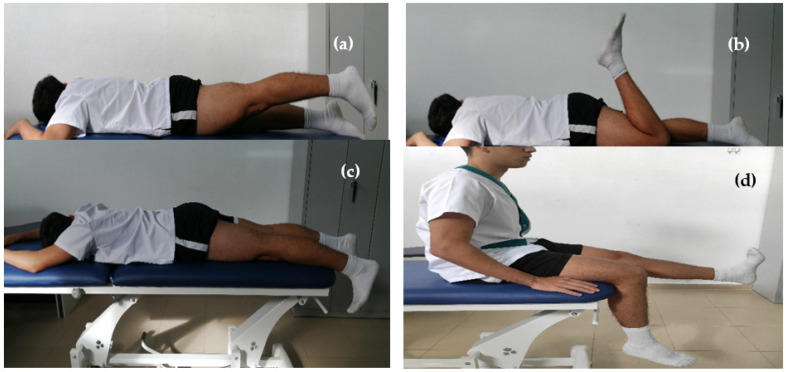
Pre-activity routine through exercise components. (**a**) leg extension in isometric; (**b**) active mobilization with knee flexion and extension; (**c**) neurodynamic tensioner with ankle dorsiflexion; (**d**) neurodynamic slider with active knee extension.

**Figure 3 ijerph-20-00744-f003:**
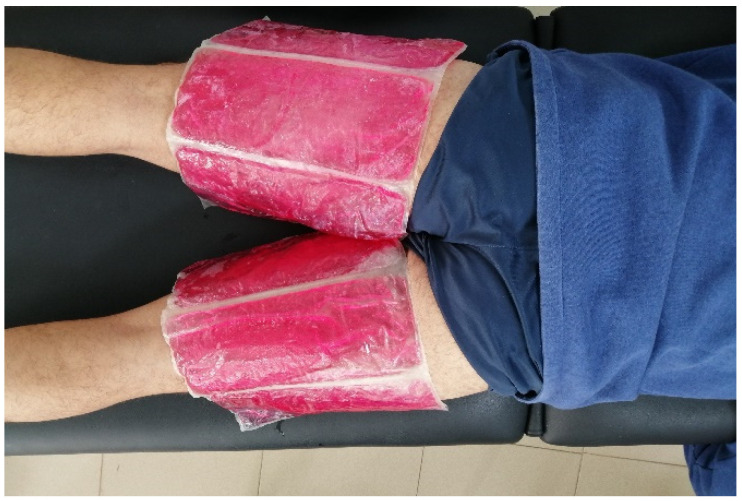
Static warm-up method with standard-size hot packs.

**Table 1 ijerph-20-00744-t001:** Characteristics of the participants.

	Total (n = 62)	SW (n = 31)	DW (n = 31)	*p*-Value
**Age**	21.1 ± 2.03; 21 [3]	21.1 ± 2.26; 21 [4]	21.1 ± 2.12; 21 [3]	0.853 *
**Gender (Male/Female)**	52 (84%)/10 (18%)	25 (81%)/6 (19%)	27 (87%)/4 (13%)	0.49 †
**Lower limb dominance (right/left)**	59(95%)/3 (5%)	29 (94%)/2 (6%)	30 (97%)/1 (3%)	0.96 †
**Height (cm)**	176 ± 0.09; 175 [0.08]	173 ± 0,07; 173 [0.07]	178 ± 0.10; 176 [0.15]	0.189 *
**Weight (kg)**	75.91 ± 17.04; 73.5 [15.50]	77.50 ± 21.33; 71 [13.50]	74.32 ± 11.60; 76 [12.50]	0.648 *
**BMI (kg/cm^2^)**	24.56 ± 5.06; 23.72 [2.75]	25.69 ± 6.63; 23.48 [4.87]	23.43 ± 2.43; 23.75 [2.62]	0.339 *
**RoM (°)**	44.36 ± 8.51; 45.9 [10.13]	45.33 ± 8.96; 46.84 [10.89]	43.38 ± 8.0; 45.32 [9.59]	0.371 *
**Stretching Tolerance (VAS 0–100 mm)**	51.8 ± 15.2; 55 [2]	50 ± 16.9; 50 [20]	53.5 ± 13.3; 60 [20]	0.363 *
**PPT (kg/cm^2^)**	5.25 ± 1.71; 5.22 [1.97]	5.6 ± 1.9; 5.5 [1.95]	4.9 ± 1.36; 4.75 [2.10]	0.116 *
**JPS (°)**	6.7 ± 3.73; 6.88 [5.78]	6.8 ± 4.2; 6.66 [5.74]	6.7 ± 3.2; 7.3 [5.9]	0.899 *
**Sport practice (h/week)**	9.71 ± 3.92; 8 [6]	9.33 ± 3.88; 8 [6]	10.10 ± 4.02; 10 [7]	0.517 *
**Sport Discipline** Running (%)	26/62 (42%)	12/26 (46%)	14/26 (54%)	0.89 †
Cycling (%)	16/62 (26%)	9/16 (56%)	7/16 (44%)	0.78 †
Football (%)	18/62 (29%)	8/18 (45%)	10/18 (55%)	0.98 †
Tennis (%)	2/62 (3%)	1/2 (3%)	1/2 (3%)	0.99 †

SW: static warm-up; DW: dynamic warm-up; BMI: body mass index; h/week: hours/week; mm: millimeters; kg: kilograms; cm: centimeters; °: degrees; %: percentages mean ± standard deviation; median (interquartile range). * *p*-value for t student test and † *p*-value for chi-square test.

**Table 2 ijerph-20-00744-t002:** Mean and standard deviation (SD) of static and dynamic warm-up interventions. Comparisons within the groups at different moments.

Group				Baseline–Immediately After	Immediately After–10 min Follow-Up	Baseline–10 min Follow-Up
	Baseline	Immediately After	10 min Follow-Up	Within-Group Mean Changes	Within-Group Mean Changes	Within-Group Mean Changes
**RoM (°)**						
**SW**	45.3 ± 8.96	46.9 ± 9.67	47.4 ± 9.41	1.6 [0.2–3.4] (−)	0.5 [−1.6–0.8] (−)	2.1 [0.4–3.7] * (0.23)
**DW**	43.4 ± 8.01	44.8 ± 9.2	43.7 ± 7.40	1.4 [0.4–3.3] (−)	1.1 [0.1–2.3] (−)	0.3 [−2.4–1.8] (−)
**Stretching Tolerance (VAS 0–100 mm)**						
**SW**	50.0 ± 16.9	36.1 ± 17.0	42.6 ± 18.4	13.9 [8.9–18.8] ** (0.82)	6.5 [1.8–11.0] * (0.36)	7.4 [2.3–12.5] * (0.41)
**DW**	53.5 ± 13.3	38.7 ± 19.1	44.8 ± 16.7	14.8 [9.4–20.2] ** (0.9)	6.1 [1.8–10.4] * (0.34)	8.7 [3.2–14.1] * (0.57)
**PPT (kg/cm^2^)**						
**SW**	5.6 ± 1.9	6.7 ± 1.81	6.2 ± 1.84	1.1 [0.6–1.5] ** (0.6)	0.5 [0.08–0.9] * (0.27)	0.6 [0.1–1.1] * (0.32)
**DW**	4.9 ± 1.36	6.6 ± 1.95	5.9 ± 1.40	1.7 [1.2–2.1] ** (1.01)	0.7 [0.1–1.3] * (0.41)	1 [0.5–1.4] ** (0.72)
**JPS (°)**						
**SW**	6.8 ± 4.22	5.5 ± 2.87	5.8 ± 3.64	1.3 [−0.5–3.0] (−)	0.3 [−1.6–1.0] (−)	1 [0.6–2.6] (−)
**DW**	6.7 ± 3.2	5.2 ± 2.3	5.7 ± 2.73	−1.5 [−0.3–2.6] * (0.53)	0.5 [−1.6–0.6] (−)	1 [0.4–2.3] (−)

RoM: range of movement; VAS: visual analogue scale; PPT: pressure–pain threshold; JPS: joint position sense. Data are reported as the mean ± SD or 95% confidence level, and the effect size as (d’ Cohen). * Indicates statistically significant within-group differences (*p* < 0.05); ** Indicates statistically significant within-group differences (*p* < 0.001).

**Table 3 ijerph-20-00744-t003:** Mean and standard deviation (SD) static and dynamic warm-up intervention. Comparisons between different moments.

		Immediately After	Mean Differences Between-Groups	10 min Follow-Up	Mean Differences Between-Groups
**RoM (°)**	SW	46.9 ± 9.67	2.1 [−2.4–6.7]	47.4 ± 9.41	3.7 [0.6–7.9]
	DW	44.8 ± 9.2		43.7 ± 7.40	
**Stretching Tolerance (VAS 0–100 mm)**	SW	36.1 ± 17.0	2.6 [−11.7–6.6]	42.6 ± 18.4	2.2 [−11.2–6.7]
	DW	38.7 ± 19.1		44.8 ± 16.7	
**PPT (kg/cm^2^)**	SW	6.7 ± 1.81	0.1 [−0.8–1.0]	6.2 ± 1.84	0.3 [−0.5–1.1]
	DW	6.6 ± 1.95		5.9 ± 1.40	
**JPS (°)**	SW	5.5 ± 2.87	0.3 [−1.05–1.6]	5.8 ± 3.64	0.1 [−1.5–1.7]
	DW	5.2 ± 2.3		5.7 ± 2.73	

RoM: range of movement; VAS: visual analogue scale; PPT: pressure–pain threshold; JPS: joint position sense. Data are reported as the mean ± SD, or 95% confidence level.

## Data Availability

Not applicable.

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
