# Peer review of "The Clinical Benefits of a Dynamic vs. Static Component as Part of a Comprehensive Warm-Up for Recreational Sports Players with Clinical Histories of Hamstring Injuries: A Randomized Clinical Trial"

_ijerph, 2022, doi:10.3390/ijerph20010744_

Round 1
Reviewer 1 Report
Formally, the work is written without reservations. The methodology is presented exhaustively. A very well conducted discussion. I have no comments on the given text.
Although personally, I'm not sure if the static warm-up is comparable to the dynamic warm-up. Locally perhaps, but globally many aspects are missing. The authors are aware of the limitations of the research and do not jump to conclusions and show the need for further research.
Author Response
In relation to the manuscript with Title “Clinical Benefit of the Dynamic vs. Static Component as Part of a Comprehensive Warm-Up on Recreational Sport Players with Clinical History of Hamstring Injury: A Randomized Clinical Trial” we would like to thank the reviewer for the comments on our manuscript. The authors would like to express our gratitude to the Reviewers for the positive feedback and useful comments. Although no comments are given by R1, as the conclusion section was ticked as “can be improved”, and it is also in line with a suggestion on reviewer 2, we have updated this section for a better understanding, trying to better link it with the results section. For sure the aspects that have been changed due to the reviewer’s comments will help improve the understanding of our research and its impact.

Reviewer 2 Report
Dear Authors
I recognize the importance of your research in the sports rehabilitation/retraining field. However, you did not let the research background clear, not showing what exactly is the level of novelty in this research. Similarly, the discussion was not enjoyed in its maximal potential for outcomes explanations. I consider your paper as a potential work for publication in IJERPH, but, several improvements are needed to reach the necessary level of quality. Follow my considerations along the pdf document, to improve and reorganize the paper. After this, resubmit the manuscript, please. Best Regards

Author Response
In relation to the manuscript with Title “Clinical Benefit of the Dynamic vs. Static Component as Part of a Comprehensive Warm-Up on Recreational Sport Players with Clinical History of Hamstring Injury: A Randomized Clinical Trial” we would like to thank the reviewer for the comments on our manuscript. For sure the aspects that have been changed due to the reviewer’s comments will help improve the understanding of our research and its impact. Each comment has been answered individually in the separate letter attached below, and changes in the manuscript have been addressed in track changes so that they can be easily identified and reviewed.
